# Topologically cloaked magnetic colloidal transport

Anna M. E. B. Rossi [1], Thomas Märker[1], Nex C. X. Stuhlmüller [1], Piotr Kuświk [2], Feliks Stobiecki[2], Maciej Urbaniak [2], Sapida Akhundzada [3], Arne J. Vereijken[3], Arno Ehresmann [3], Daniel de las Heras [1] & Thomas M. Fischer [1] ✉

Cloaking is a method of making obstacles undetectable. Here we cloak unit cells of a magnetic pattern squeezed into an otherwise periodic pattern from a magnetically driven colloidal flow. We apply a time-periodic external magnetic field loop to an ensemble of paramagnetic colloidal particles on the deformed periodic magnetic pattern. There exist topological loops where the particles avoid to trespass the cloaked regions by robustly traveling around the cloak. Afterwards the ensemble of particles continues with a motion identical to the motion as if the distorted region were nonexistent and the ensemble would have trespassed the undeformed region. We construct the cloak by continuously squeezing new conformally mapped unit cells between those of the originally undeformed and periodic pattern. We find a cloaking/decloaking transition as a function of the size and shape of the newly squeezed-in region. A cloak is scalable to arbitrary size if the biholomorphic map from the undistorted periodic lattice to the region outside the cloak locally rotates by less than an angle of forty five degrees. The work generalizes cloaking from waves toward particles.

The scattering of waves or particles is one of the major techniques employed to study the properties of scatterers. We learn about the internal static and dynamic structure of the scatterer by measuring the dynamic structure factor, i.e., the likelihood of a transfer of momentum and energy past the scattering event as compared to the situation prior to the encounter. Of course we will not learn anything about the scatterer if the waves or particles probing our scatterer do not interact with the scatterer. In such a situation the probing waves or particles merely pass past the scatterer without recognizing its existence. A similar situation occurs when the probe interacts with the scatterer in such a peculiar way that the probe's space-time dynamics is altered only intermittently but remain unchanged asymptotically so the scatterer hoodwinks us by pretending to be nonexistent due to a lack of alteration of the asymptotic space-time behavior of the probe. We then say the scatterer is cloaked.

There have been theoretical suggestions[1] and experimental verifications[2] on how to cloak an object from electromagnetic waves[3], be it in the microwave[4], terahertz[5], or the visible[6–13] regime. One has cloaked against acoustic waves[14–16] as well as elastic deformations[17], phonons[18,19], and water waves[20,21]. The cloaking often requires the use of a meta material[22] that shows a time-reversed response[23] of the probing waves. Supersymmetric quantum mechanics with an algebra that produces a hierachy of nonscattering potentials[24,25] can cloak the asymptotic observation of the potential and generalize the concept of cloaking from waves via matter waves[26] toward particles[27].

In previous work, we have induced topologically non-trivial transport of colloidal[28–37] and of macroscopic[38,39] magnetic particle systems that are subject to dissipative dynamics by applying topological nontrivial modulation loops of an external magnetic field. In this work, we cloak a selected region of a 2D magnetic periodic pattern from this topological transport, by deforming the pattern

[1]Institute of Physics, Universität Bayreuth, Bayreuth, Germany. [2]Institute of Molecular Physics, Polish Academy of Sciences, Poznań, Poland. [3]Institute of Physics and Center for Interdisciplinary Nanostructure Science and Technology (CINSaT), University of Kassel, Kassel, Germany. ✉e-mail: Thomas.Fischer@uni-bayreuth.de

around the region that is chosen to be cloaked. Instead of trespassing the cloaked region, the colloidal ensemble splits in two and moves around its border, called the cloak, and afterward rejoins and catches up with the same space-time dynamics as if the cloaked region were absent.

From a fundamental point of view, the work generalizes the concept of cloaking to dissipative (non-Hamiltonian) particle systems. The colloidal particles, albeit bombarded with fluctuating forces from the solvent molecules, are forced back to the topologically protected path they would have followed without the cloak. From an application point of view, our system might serve as a security ward for chemicals on a lab-on-the-chip device. The security ward would be unreachable for other chemicals transported on the chip. This is just one example of the use of cloaking for the transport of colloids. There are abundant applications of cloaking in wave-like systems[1–26]. Since we show here

that particle-like systems can also exhibit cloaking, this opens possibilities for new applications in particle-related fields.

## Results and discussion

Paramagnetic colloidal particles (diameter 1 and 2.8 μm) immersed in water are placed on top of a two-dimensional magnetic pattern that is covered with a 1 μm thick polymer film. We call the plane the particles move in the action space $\mathcal{A}$. The pattern depicted in Fig. 1 is a square lattice of alternating regions with positive (gray) and negative (olive/green) magnetization (red arrows) relative to the direction normal to the pattern. A uniform time-dependent external magnetic field of constant magnitude is superimposed to the inhomogeneous time-independent magnetic field generated by the pattern. The orientation of the external field of constant magnitude changes adiabatically along a closed loop $\mathcal{L}_\mathcal{C}$ in control space $\mathcal{C}$ (a sphere of radius $H_{ext}$, see Fig. 1a). There are special orientations in

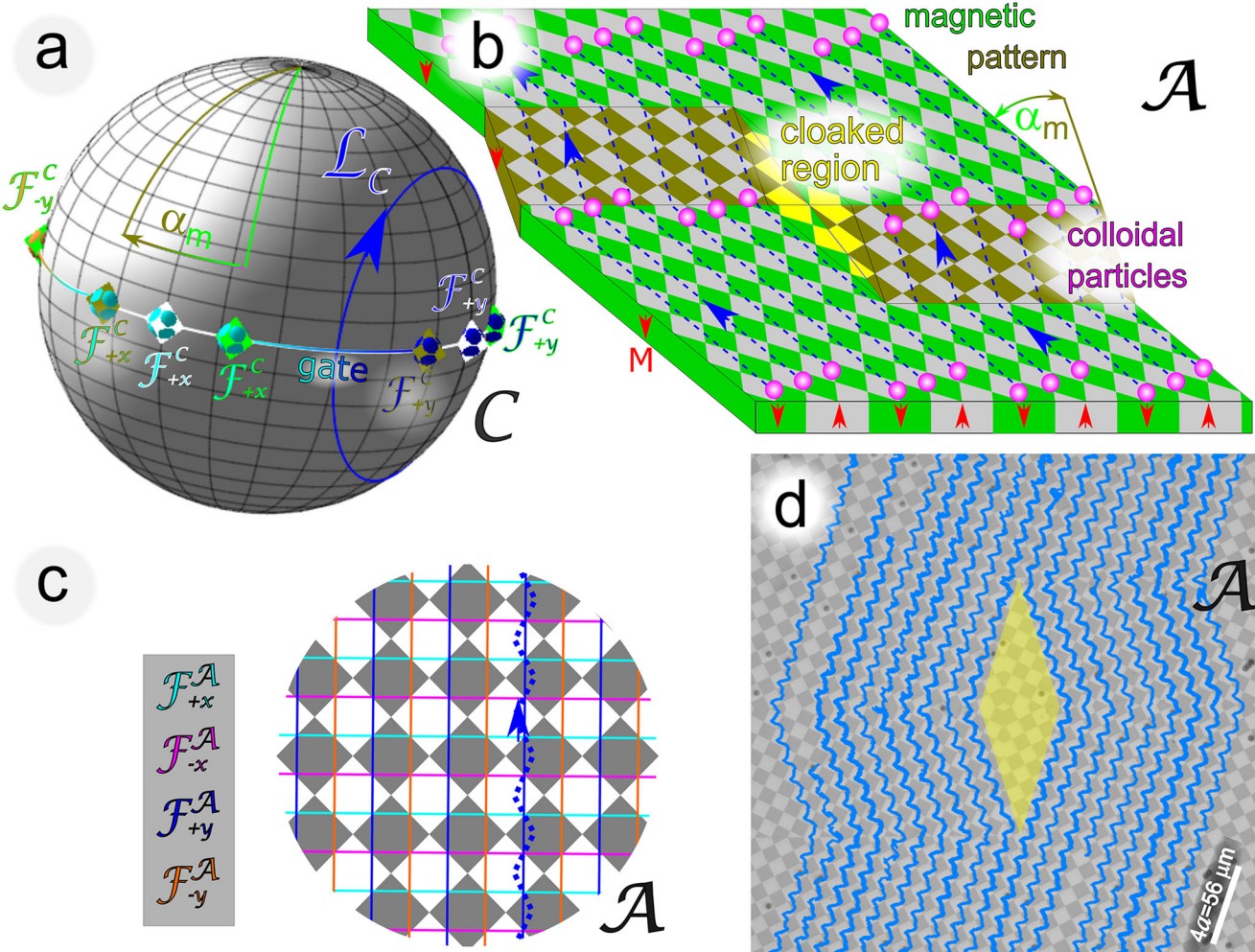

**Fig. 1 | Cloaking by a twin boundary pattern. a** We apply an external magnetic field of constant magnitude that varies as a function of time along a modulation loop $\mathcal{L}_\mathcal{C}$ (blue) in control space $\mathcal{C}$. **b** The external magnetic field penetrates our cloak pattern consisting of up (gray/yellow) and down (green/olive) magnetized domains. The domains generate a second, time-independent, heterogeneous, magnetic field above the pattern. The olive (green) square pattern is rotated clockwise (counterclockwise) by half the magic angle $\pm \alpha_m/2 = \pm \arctan(1/3)$. The counter-rotation generates twin boundaries between both patterns along the primitive unit vectors of the unrotated pattern. The modulation loop of the external magnetic field in (**a**) winds around the blue $\mathcal{F}^\mathcal{C}_{+y}$-fence points of the unrotated (white diamond in panel **a**, white pattern in panel **c**) and rotated (olive and green) patterns and, therefore, transports the paramagnetic colloids along the local rotated $\mathcal{F}^\mathcal{A}_{+y}$-fence directions in action space $\mathcal{A}$. Hence, these fence lines travel

around the yellow-cloaked region. A detailed picture of the particle trajectories is shown for the unrotated pattern in (**c**) together with the various $\mathcal{F}^\mathcal{A}_{\pm x, \pm y}$- fences. A particle takes a slalom-like path (dashed blue line) along one of the blue $\mathcal{F}^\mathcal{A}_{+y}$-fences corresponding to the blue sphere and white diamond $\mathcal{F}^\mathcal{C}_{+y}$-fence point in control space. The particle switches sides at the crossings of two $\mathcal{F}^\mathcal{A}$-lines whenever the control loop $\mathcal{L}_\mathcal{C}$ switches from the northern to the southern (southern to the northern) hemisphere through the gate between the corresponding fence points in $\mathcal{C}$. The final and initial conformation of the colloids before and after passing by the cloaked region are the same. **d** Microscopy image of the colloids with experimental trajectories of an ensemble of colloids traveling around the cloak. Supplementary video 1 shows the dynamics of the colloidal ensemble traveling around the diamond-shaped cloak.

control space, called fence points. These orientations are the only orientations for which the position of a paramagnetic particle inside one unit cell is marginally stable. We have marked the fence points $\mathcal{F}^{\mathcal{C}}_{\pm x,\, \pm y}$ by a superposition of colored diamonds and colored spheres in Fig. 1a. The color of the diamonds codes the two rotated and one unrotated orientation of the pattern. For one orientation of the pattern, the color of the sphere codes the type $\pm x$, $\pm y$ of four different fence points. In action space, Fig. 1c, these fence points generate fence lines $\mathcal{F}^{\mathcal{A}}_{\pm x,\, \pm y}$ of these same four colors. In references[28–30] we have shown that per winding of the control loop $\mathcal{L}_{\mathcal{C}}$ around a fence point $\mathcal{F}^{\mathcal{C}}_{\pm x,\, \pm y}$ in control space $\mathcal{C}$ the colloids are transported by one unit cell on a slalom-like path along the corresponding $\mathcal{F}^{\mathcal{A}}_{\pm x,\, \pm y}$-lines in action space $\mathcal{A}$. In the white and gray non-rotated square pattern of Fig. 1c) the blue loop of Fig. 1a causes the paramagnetic particles to move along the blue path in the pattern. The motion is topologically protected, which means that if we change the loop, e.g. by rotating the loop by a sufficiently small angle around the equator, the motion over a period does not change provided that the winding numbers around the (white diamond) fence point do not change. Similarly, if we rotate the pattern by a sufficiently small angle its orientation change will also just rotate the transport direction with the pattern. In the pattern outlined in Fig. 1b, we use this to cloak the yellow-shaded diamond-shaped region by rotating the square pattern by half a magic angle $\alpha_m/2 = \pm \arctan(1/3)$[40] into opposite directions (olive and green). Due to the magic angle, the new pattern consists of a green and an olive twin crystal joined at a twin boundary.

With respect to the fence points of the unrotated pattern (white diamonds), the fence points of the crystal twins (olive and green diamonds) in the control space, see Fig. 1a, are rotated by the same amount around the equator without affecting the winding number of the blue $\mathcal{L}_{\mathcal{C}}$-loop around the corresponding blue $\mathcal{F}_{\mathcal{C}}$-fence. The path of the colloids, therefore, first follows the locally rotated fence direction on the green twin, then twists away from the green direction by the magic angle when entering the olive twin and returns to the original direction after moving on to the next green twin. Colloidal particles, therefore, travel around the yellow-cloaked diamond-shaped region. The conformation of an ensemble of particles is fully recovered after passing the olive twin. From the final conformation, we, therefore, cannot tell whether or not the ensemble has encountered an obstacle in its way. In Fig. 1d we show a microscope image of the colloids together with measured trajectories of individual colloidal particles in the ensemble. The particles follow the locally topologically protected travel direction avoiding the cloak. Supplementary video 1 shows the dynamics of the colloidal ensemble traveling around the diamond-shaped cloak. Some particles in the video have assembled into colloidal bipeds[33], which are several colloidal particles forming a rod. Bipeds of two particles are topologically equivalent to single colloidal particles and are thus transported the same way. Some colloidal particles sediment into the cloak from above, but note that there is no 2D transport of colloidal particles into the cloak.

In Fig. 1 we have used two global rotations that rotate the entire pattern of Fig. 1c either toward the green or the olive pattern of Fig. 1b. The two global rotations cause the formation of sharp twin boundaries (between the green and the olive patterns) that must be passed by the particles and that are the source of more complicated transport. We can avoid such domain walls using local instead of global rotations. A local rotation field preserves the shape of the pattern locally but smoothly rotates neighboring regions relative to each other. That is, we apply a conformal mapping[41], $\mathbf{M}_u(\mathbf{u}(\mathbf{z})) \to \mathbf{M}_c(\mathbf{c}(\mathbf{z}))$ from the periodic magnetization $\mathbf{M}_u$ in the undeformed $u$-plane to a magnetization $\mathbf{M}_c$ on the cloaked pattern in the $c$-plane. The magnetization on our deformed pattern at the location $\mathbf{c}(z)$ is the same as the magnetization at the location $\mathbf{u}(z)$ on the undeformed pattern. Both locations $\mathbf{u}(z)$ and

$\mathbf{c}(z)$ are parametrized by the complex variable $z$. The cloaked magnetization at position $\mathbf{c}(z)$ reads:

$$\mathbf{M}_c(\mathbf{c}(z)) = M_z \mathbf{e}_z \, \mathrm{sign}\left( \sum_{i=1}^{2} \cos[\mathbf{k}_i \cdot \mathbf{u}(z) + \psi_i] \right), \qquad (1)$$

where the $\mathbf{k}_i$ are the primitive reciprocal unit vectors of the undistorted lattice and the phases $\psi_i$ adjust the shift of the Wigner Seitz cell center of the periodic pattern with respect to the origin $\mathbf{u} = \mathbf{0}$. The vectors $\mathbf{u}(z) = [\Re u(z), \Im u(z)]$ and $\mathbf{c}(z) = [\Re c(z), \Im c(z)]$ span the complex $u$- and $c$-planes of the undeformed and conformally deformed lattice and are described by the undeformed complex variable $u(z)$ respectively the conformally deformed complex variable $c(z)$ that are both parametrized by the complex parameter $z$. For the parametrization of the undeformed plane we chose the 2 : 1-parametrization

$$u(z) = z + R^2/z \qquad (2)$$

such that $u(z) = u(R^2/z)$, i.e., each $u \in \mathbb{C}$ has two preimages in the $z$-parameter-plane, one with $|z| < |R|$ and one with $|z| > |R|$. Both parameter regions are glued together at $|z| = |R|$. This parameter circle is mapped onto the cloak boundary in the undeformed and the deformed plane. The cloak is folded to a cut between $u = -2R$ and $u = 2R$ in the undeformed plane where either side of the upper pattern $\mathbf{M}^{>}_u = \mathbf{M}_u(|z| > |R|)$ is glued to the other side of its identical symmetric lower partner pattern $\mathbf{M}^{<}_u = \mathbf{M}_u(|z| < |R|)$. The argument $\arg(R)$ thus gives us the cut-direction, $\arg(R) = \pi/2,\ \pi/2,\ \pi/4$ in Fig. 2a, b, and in Fig. 3. Mathematically, we move from the upper toward the lower pattern when crossing the cloak. By aligning the reciprocal unit vector $\mathbf{k}_y$ parallel to the cloak and by a proper choice of the pattern phase $\psi_y$ we may place an undeformed $\mathcal{F}^{\mathcal{A}}_{-y}$ fence line right onto the cloak. Our control loop $\mathcal{L}_{\mathcal{C}}$ around the $\mathcal{F}^{\mathcal{C}}_{+y}$ fence point lets the particles move along the undeformed $\mathcal{F}^{\mathcal{A}}_{+y}$-fence lines without coming close to the cloak and the mathematical construction of using two identical patterns remains without effect. In fact, the symmetric lower pattern $\mathbf{M}^{<}_u$ remains completely hidden.

We now move to the behavior in the deformed $c(z)$-plane. Since our transport is topologically robust, we may squeeze the cloak open like in Fig. 2 by choosing a different parametrization $c(z) \neq u(z)$ that is biholomorphic for $|z| > |R|$ of the deformed pattern, which removes the pattern $\mathbf{M}^{>}_c$ from the region inside the cloak and refills the cloak with deformed unit cells (yellow) of the lower symmetric partner pattern $\mathbf{M}^{<}_c$. The lower symmetric partner pattern, albeit now producing a real magnetic field, nevertheless remains irrelevant for the transport with the $\mathcal{L}_{\mathcal{C}}$ loop. The transport is topologically protected from such a deformation if the local rotation of the conformally mapped upper pattern $\mathbf{M}^{>}_c$ with respect to the undeformed pattern is sufficiently small, $|\arg[dc(z)/du(z)]| < \pi/4$.

For the conformally mapped plane, we present three choices. The first choice $u(z)$, $c_{\bigcirc}(z) = z$ is a conformal map around a circular cloak, see the sketch in Fig. 2a, the second choice $u(z)$, $c_0(z)$ is a conformal map around a boat like a cloak with two cusps at the front and at the tail, see Fig. 2b, and the last map $u(z)$, $c_{\diamond}(z)$ is around a square cloak, see Fig. 3d and e. All three conformal distortions squeeze the original periodic pattern outside the cloak by wrapping it around our cloaked yellow lower symmetric partner pattern. The lower symmetric partner pattern is the analytical continuation of the upper pattern into the cloak. Since the deformed unit cells inside the cloak were hidden in the undistorted pattern, they are not covered with deformed trajectories of the colloidal ensemble.

The original pattern splits at two branch points where $du/dz = 0$. These points are placed in the front and the rear of the yellow region. Regions outside of the cloak in Fig. 2a, where the conformal map locally rotates the pattern by more than $\pi/4$ are colored in red. Figure 2a also shows the positions of a few members of the two families

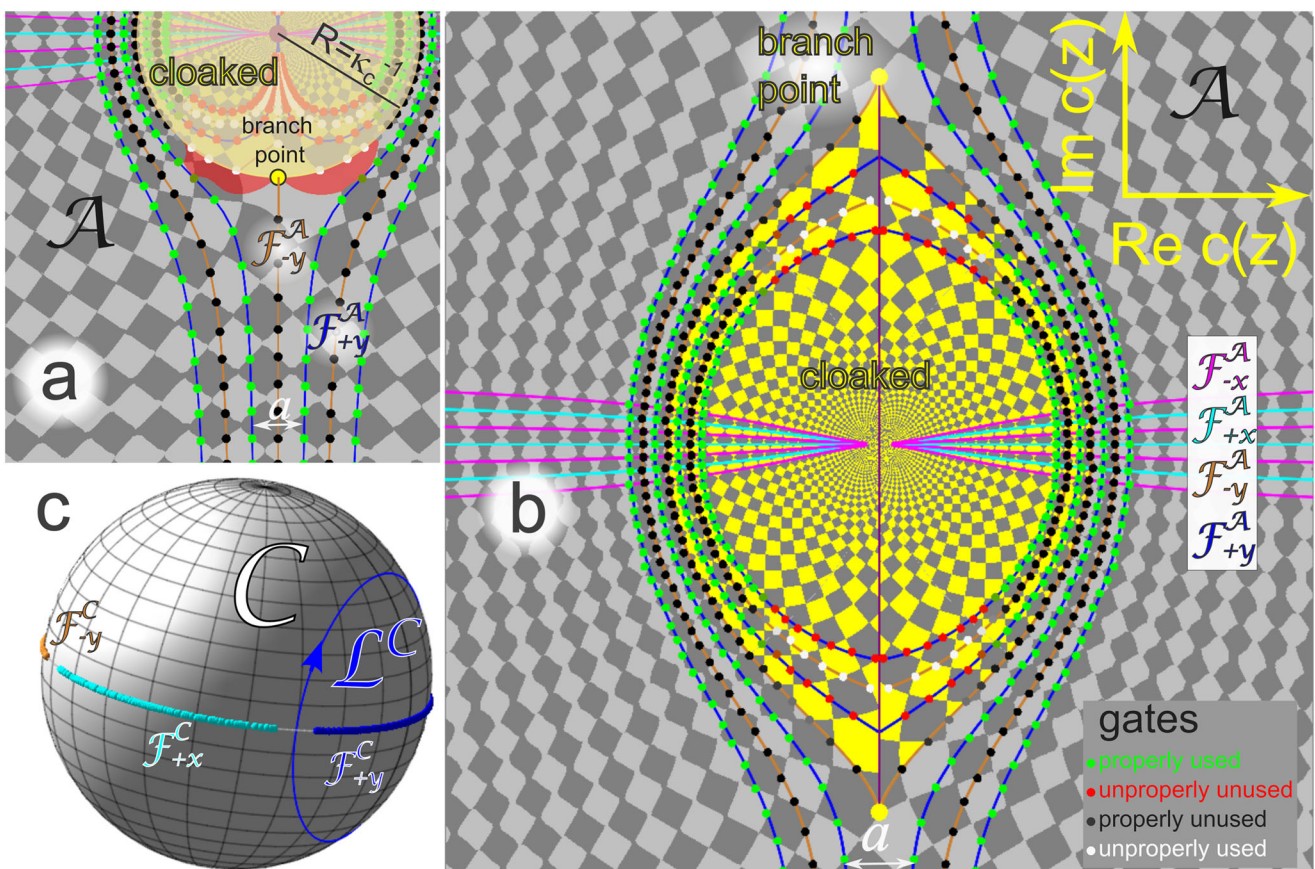

**Fig. 2 | Cloaking with conformal maps. a** Scheme of a conformal map in action space near the branch point. The original $\mathcal{F}^{\mathcal{A}}_{-y}$-fence running over the cut is split into two branches near the branch point after the map is applied and travels around the yellow-cloaked region. Due to the branch point and the finite radius of curvature of the border at the branch point, there are two red regions adjacent to the branch point where the local rotation of the pattern $|\arg[dc(z)/du(z)]| > \pi/4$ is too large such that fence points in control space corresponding to this region are no longer encircled by the control loop. If one wants to use the blue $\mathcal{F}^{\mathcal{A}}_{+y}$-fence lines for transport, this sets a lower bound $\kappa > \kappa_c$ for the curvature of the cloak border at the branch point. **b** Conformally mapped and analytically continued (yellow) cloak pattern according to map $c_0(z)$ together with some of the conformally mapped fence lines $\mathcal{F}^{\mathcal{A}}_{\pm x, \pm y}$ and gates of the original pattern. The cloak shape works for arbitrary large sizes. **c** Control space $\mathcal{C}$ of the cloaked pattern in b) with the rotated fence points $\mathcal{F}^{\mathcal{C}}_{\pm x, \pm y}$ outside the cloak. Fence points of the four different fence families are well separated from each other, allowing the control loop to wind around one family without excluding any family member nor including any member of a different fence family. Supplementary video 2 shows the opening of the boat-shaped cloak and the associated broadening of the fence points in control space as we increase the cloak size $R$.

of the conformally mapped fence lines $\mathcal{F}^{\mathcal{A}}_{\pm y}$ (orange, blue) that are the deformed marginally stable positions of the particles when the field is perpendicular to the $\pm y$-direction. Wherever the tangent vector to the deformed fence lines is at an angle less than $\pi/4$ to the undeformed fence lines, the loop $\mathcal{L}_{\mathcal{C}}$ around all deformed fence points in control space will let the particles follow the deformed fence lines and thus travel around the cloaked yellow analytically continued region.

The central orange fence in Fig. 2a is part of the $\mathcal{F}^{\mathcal{A}}_{-y}$-fences. It turns by $\pm\pi/2$ (i.e., more than $\pi/4$) at the branch point and passes through the red regions. A loop around the $\mathcal{F}^{\mathcal{C}}_{-y}$-fence in the control space will not transport the particles along this line. A loop around the $\mathcal{F}^{\mathcal{C}}_{+y}$ fence in control space, however, works and transports the particles along the blue $\mathcal{F}^{\mathcal{A}}_{+y}$-fences if the curvature $\kappa$ of the cloak near the branch points is large enough ($\kappa > \kappa_c = 1/(\pi a)$), such that the red region is avoided by all $\mathcal{F}^{\mathcal{A}}_{+y}$-fences.

For a curvature at the branch point smaller than the critical curvature $\kappa < \kappa_c$, the blue $\mathcal{F}^{\mathcal{A}}_{+y}$-fence closest to the yellow region cuts through the red region. The fence points $\mathcal{F}^{\mathcal{C}}_{+y}$ in control space corresponding to the red region will no longer be circulated by the control loop $\mathcal{L}_{\mathcal{C}}$ since they are rotated out of it. The topological protection of the path is lost, and the shape is decloaked. Our second choice of cloak, Fig. 2b has a cusp at both branch points such that the rotation of the conformal map is small

enough $|\arg[dc(z)/dr(z)]| < \pi/4$ in the entire region $|z| > |R|$ outside the cloak.

The crossings of a $\mathcal{F}^{\mathcal{A}}_{\pm x}$-fence with a $\mathcal{F}^{\mathcal{A}}_{\pm y}$-fence is a gate $g_{\mathcal{A}}$. Gates in Fig. 2a and b are marked as circles. The particles cross the gates when the external field passes the equator between the corresponding deformed fences in the control space. When cloaking works, Fig. 2b, all gates on the orange $\mathcal{F}^{\mathcal{A}}_{-y}$-fences outside the cloak are not used (black circles), and all gates on the blue $\mathcal{F}^{\mathcal{A}}_{+y}$-fence are used (green circles). For cloaking to work outside the cloak, there must be no unproperly used gates on the $\mathcal{F}^{\mathcal{A}}_{-y}$ fence (white circles in Fig. 2) nor unproperly unused gates on the $\mathcal{F}^{\mathcal{A}}_{+y}$ fence (red circles in Fig. 2b). Unproper gates are, however, allowed inside the cloak. When we plot the fence points $\mathcal{F}^{\mathcal{C}}_{\pm x, \pm y}$ in control space, Fig. 2c, corresponding to all the fence crossings in action space outside the cloak, Fig. 2b, we find all four different fence families to be well separated from each other, allowing the control loop $\mathcal{L}_{\mathcal{C}}$ to wind around only the $\mathcal{F}^{\mathcal{C}}_{+y}$ family without excluding any family member nor including any member of a different fence family. Supplementary video 2 provides a visualization of the opening of the cloak and the corresponding broadening of the fence families.

In Fig. 3a we experimentally verify the cloaking using a colloidal ensemble above a circular cloak pattern with a cloak of curvature $a^{-1} = \kappa > \kappa_c$. All members of the ensemble respect the

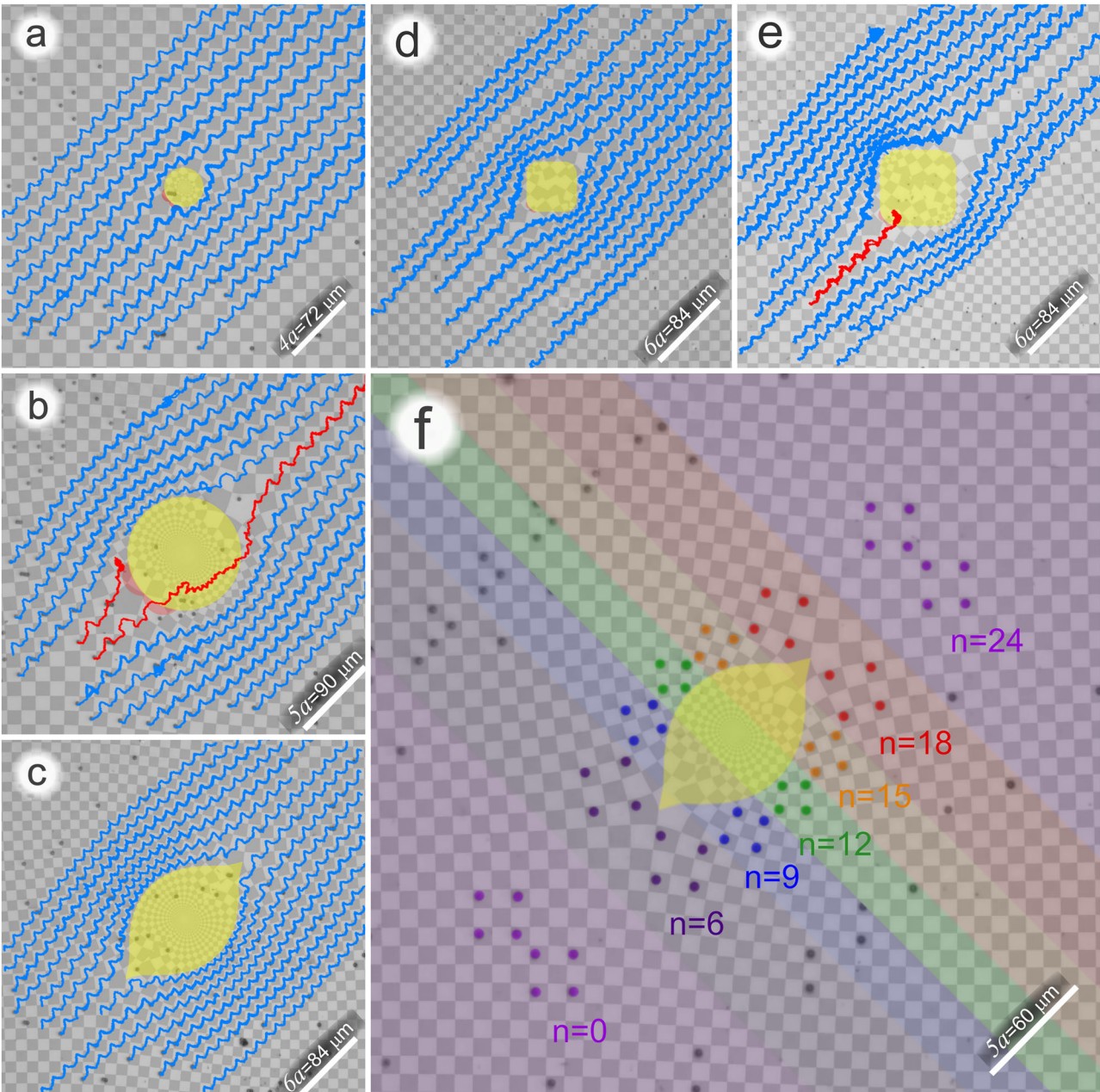

**Fig. 3 | Scalable and nonscalable cloaks. a** Succesfull cloaking of a circular cloak of curvature $a^{-1} = \kappa > \kappa_c$ showing experimentally tracked trajectories of a colloidal ensemble. **b** Decloaking of a circular cloak of curvature $(4a)^{-1} = \kappa < \kappa_c$ showing experimentally tracked trajectories of a colloidal ensemble. The trajectories of two colloidal particles of the ensemble (red) do not follow the proper path and decloak the protected circular region. **c** Cloaking of a large scalable boat-shaped cloak showing experimentally tracked trajectories of a colloidal ensemble successfully traveling around the cloak. **d** Trajectories of a colloidal ensemble successfully traveling around a rounded square-shaped cloak with $|R| = 2a$. **e** Trajectories of a colloidal ensemble traveling around a rounded square-shaped cloak with $|R| = 3a$. Despite the in-principle scalable shape of a square, the rounded square is decloaked by one colloidal particle of the colloidal ensemble penetrating into the square. Supplementary videos 3–5 show the dynamics of the colloidal ensemble traveling

around the circular-, the boat- and the square-shaped cloaks of various sizes. **f** Overlay of microscope images showing the conformation of an ensemble of eight colloidal particles after $n = 0, 6, 9, 12, 15, 18, 24$ repetitions of the loop $\mathcal{L}_{\mathcal{C}}$. The different colored areas correspond to the video image at these times. The initial twin square conformation ($n = 0$ purple) is split into two squares after six repetitions of the loop when encountering the branch point of the boat shaped cloak. The squares are then rotated in opposite directions on either side of the cloak ($n = 9, 12, 15$), rejoin after eighteen repetitions of the loop, and the twin square conformation is completely recovered after twenty-four repetitions after having passed the cloak. Supplementary video 6 shows the dynamics of the conformation of eight colloidal particles traveling around a boat-shaped cloak. The conformation of the particles is restored after passing the cloak.

cloak and the trajectories of all individual members of the ensemble circumvent the cloak by traversing properly used gates. In Fig. 3b, we show the trajectories of a colloidal ensemble above a cloak of curvature $(4a)^{-1} = \kappa < \kappa_c$. Two trajectories (red) enter the red-shaded region in front of the cloak, where $|\arg[dc(z)/du(z)]| > \pi/4$. As a result, one particle gets stuck in the

red region, and the other particle uses the wrong gate, penetrates into the cloak, and returns to the right path. However it returns with a time delay, and thus both trajectories decloak the circular yellow region. Supplementary video 3 shows the dynamics of the colloidal ensemble traveling around the circular cloaks of various sizes.

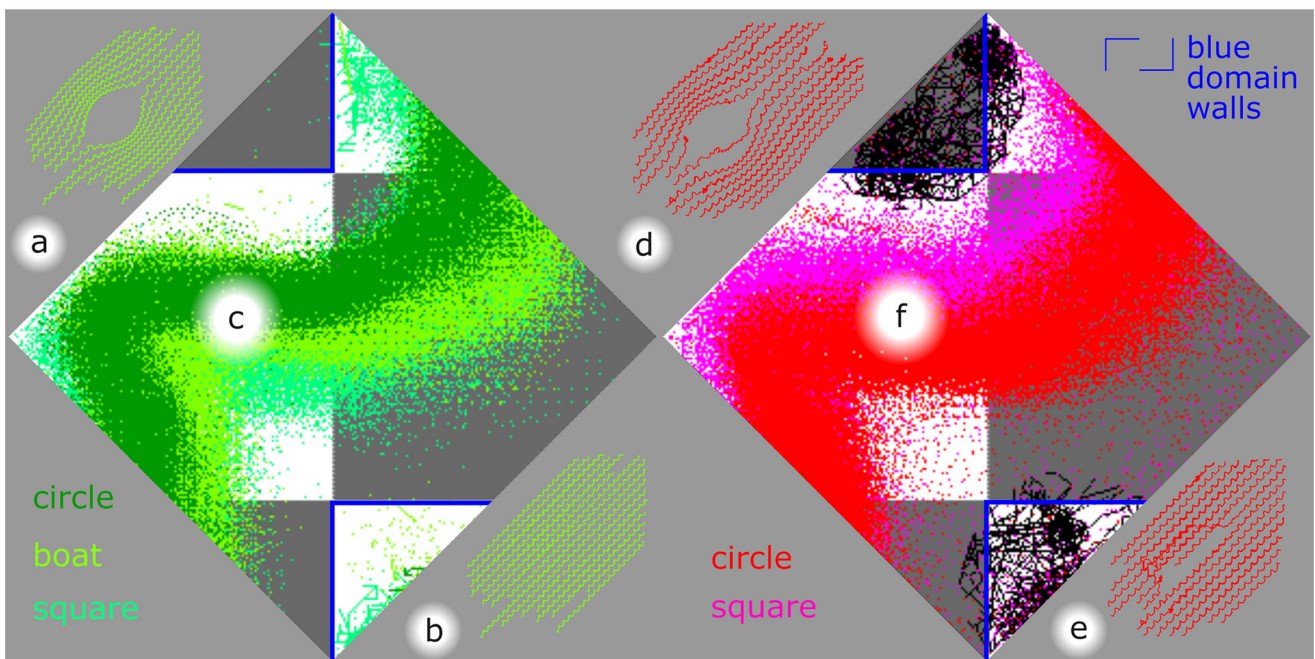

**Fig. 4 | Cloaking and decloaking. a** Trajectories of a colloidal ensemble in the complex $c$-plane traveling around a properly cloaked boat-shaped cloak. **b** The same trajectories remapped into the $u$-plane, where the pattern is periodic. **c** Mapping of the ensemble member positions of several different successfully cloaked shapes into the Wigner Seitz cell of the periodic pattern in the $u$-plane making use of the periodicity of the pattern. Trajectories are connected via lines in the region near the blue domain walls. **d** Trajectories of a colloidal ensemble in the complex $c$-plane on a decloaked circular cloak shape pattern. **e** The same trajectories remapped into the $u$-plane, where the pattern is periodic. **f** Mapping of the ensemble member positions of several different decloaked shapes into the Wigner Seitz cell of the periodic pattern in the $u$-plane making use of the periodicity of the pattern. In the region near the blue domain walls points of the trajectories are connected by black lines. Properly cloaked shapes contain no trajectories in the Wigner Seitz cell that cross the domain walls while decloaked shapes contain transfers of particles via these domain walls.

We thus find a cloaking/decloaking transition as a function of the radius of curvature of the cloak at the branch point. There are cloak shapes like the circle that are limited in size, and there are cloak shapes like the boat that can be scaled to arbitrary size without being decloaked, Fig. 3c. A scalable cloak must have infinite curvature at the two branch points. There is no cusp of the cloak shape at the branch point of the circular cloak, i.e., the unit cell at the branch point locally rotates by $\pi/2$. We achieve a cusp $(|\arg[dc(z)/du(z)]||_{z=\pm R} < \pi/4)$ at the branch point with the second map $c_\Diamond(z)$. Like the differential of the first map, the differential of the second map must vanish at the branch points $(dc/dz|_{z=\pm R} = du/dz|_{z=\pm R} = 0)$. This is the case for the boat-shaped and the square-shaped cloak. In addition, the tangent vectors to the cloak must enclose angles of less than $\pi/4$ with the cut direction. Supplementary video 4 shows the dynamics of the colloidal ensemble traveling around such a scalable boat-shaped cloak for two different boat sizes. The limiting scalable shape to be cloaked should thus be a square cloak facing the branch points at opposite corners and edges at angles $\pm \pi/4$ to the cut direction. We present a square cloak in Fig. 3d with rounded edges. The rounded edges appear when truncating the series expansion of $c_\Diamond(z)$, see the Appendix. Because of the rounding of the edges, we find that the rounded square cloak is not scalable. Figure 3d shows the trajectories of a colloidal ensemble consisting of particles of diameter $d = 1\,\mu m$ above a rounded square cloak of size $|R| = 2a$ that properly travel around the cloak. A larger cloak of size $|R| = 3a$ in Fig. 3e, however, lets one member (red trajectory) of the ensemble decloak the rounded square. Supplementary video 5 shows the dynamics of the colloidal ensemble traveling around the rounded square-shaped cloaks of various sizes. Again, the decloaking is due to the trajectories entering a region with $|\arg[dc(z)/du(z)]|| > \pi/4$. Figure 3f

shows trajectories of a colloidal ensemble above a properly cloaked boat-shaped region of a size comparable to the decloaked circular region. The absence of a region with $|\arg[dc(z)/du(z)]|| > \pi/4$ lets the ensemble pass the boat unharmed. Figure 3f presents the overlay of microscope images showing the conformation of an ensemble of eight colloidal particles after $n = 0, 6, 9, 12, 15, 18, 24$ repetitions of the loop $\mathcal{L}_\mathcal{C}$. The initial conformation (n = 0, purple) is a twin square with the colloids arranged into two (side by side) squares rotated by $\pi/4$. The conformation is split into two squares after six repetitions of the loop when encountering the branch point of the boat-shaped cloak. The two squares travel around the cloak on opposite sides. The squares are then rotated in opposite directions on either side of the cloak ($n = 9, 12, 15$), rejoin after eighteen repetitions of the loop, and the twin square conformation is completely recovered after having passed the cloak ($n = 24$, purple). Supplementary video 6 shows the dynamics of the conformation of eight colloidal particles traveling around a boat-shaped cloak. The recovered conformation of the particles also arrives at the final destination at the same time it would have arrived without the cloak standing in its way.

The sufficient requirement for a scalable conformal map, $c(z)$, is that it must be a biholomorphic map from the region outside the unit circle, $|z| > |R|$, to the region outside the cloak, $|\arg[dc(z)/du(z)]|| < \pi/4$ for $|z| > |R|$, as well as $\lim_{|z| \to \infty} c(z) = z$. There are no requirements for $c(z)$ inside the cloak, $(|z| < |R|)$.

We checked the conformal invariance of the cloaking device under these restrictions by experiments where we tracked the colloidal particles in the ensemble. Our model predicts the particles to follow a slalom-like path through the gates $g_\mathcal{A}$ of the conformally mapped $\mathcal{F}_{+y}^\mathcal{A}$-fence lines independent of the details of the loop. In Fig. 4a, we show the trajectories of the ensemble around the boat-shaped cloak. In

Fig. 4b we conformally remap these ensemble trajectories from the complex $c$-plane into the $u$-plane, where the pattern is undeformed and periodic. The periodic motion of the remapped trajectories can be nicely seen. Finally, we use the periodicity of the undeformed pattern to transport the measured particle positions into the Wigner-Seitz cell of the periodic pattern. Figure 4c shows all observed positions of colloidal particles from different shapes of cloaks that are properly cloaked but mapped into the Wigner Seitz cell of the undeformed $u$-space. One can see that the positions fall onto one master curve. Inside the Wigner Seitz cell, we mark the domain walls (blue) near the improper gates. The density of trajectory points is high in the region of the master trajectory, but low in the region that would be void if everything was perfectly cloaked. In Fig. 4c, we therefore plot trajectories as points in the high-density region and as lines in the low-density region. Points in the low-density region not connected to lines correspond to trajectories leading toward the high-density region via the Wigner Seitz cell boundaries and not via the blue domain walls. Note that most of the trajectory lines in Fig. 4c in the region close to the blue domain walls arise from the square cloak, specifically from the small unit cells near the waist of the square. No matter which shape, there are no transfers of trajectories across the marked domain walls, as can be inferred for the boat shape also from Fig. 4b.

Figure 4d–f repeat the same mapping for patterns that are decloaked by one or more members of an ensemble. Here the remapping into the $u$-plane fails to produce a periodic arrangement of trajectories around the cut in the $u$-plane, and the surroundings of the cut remain free of trajectories, Fig. 4e. Trajectories that are about to enter this region are scattered away from the cut onto positions of the lower symmetric partner pattern $\mathbf{M}^<$, which corresponds to particles entering the cloak in the $c$-plane, Fig. 4d. Using the periodicity of the pattern and mapping the trajectories into a Wigner Seitz cell of the $u$-plane, Fig. 4f, again produces the same master curve as in Fig. 4c. Like in Fig. 4c, we plot the trajectories near the master curve as points and we connect the trajectory points near the blue domain walls with black trajectory lines. Note that those lines frequently cross the blue domain walls. Such crossings are absent in Fig. 4c, emphasizing the difference of cloaked and decloaked trajectories. These crossings all arise due to the transport behavior of the red trajectories near the branch points (Fig. 3) and it shows that there is only a small region where the decloaking changes the transport behavior. Ensemble members that decloak the different shapes transfer just once during their travel via the wrong gates or the blue-marked domain walls.

The topological protection works well for the boat-shaped cloak. We observe the decloaking of the non-scalable circular shape as a function of size occurring near $|R| = \pi a$ where the scattering increases and the particles show up at the wrong gate. Of course, when we identify the misled particles, they correspond to particles in the red region of Fig. 2a.

We provide a topologically robust conformally mapped way of cloaking the transport of ensembles of paramagnetic colloidal particles around specifically designed forbidden regions. The cloak is inserted without affecting the future particle dynamics once the particles have passed around the cloaked region. The work generalizes the concept of cloaking from Hamiltonian toward dissipative particle systems. This type of cloaking specific regions could be useful for future lab-on-the-chip applications, where chemicals inside the cloak can be protected from others being transported around the cloak.

## Methods
### Cloak shapes
The three choices of conformal maps $c(z)$ are the circular cloak:

$$c_\circ(z) = z, \tag{3}$$

the boat-shaped cloak

$$c_0(z) = z\sqrt{1 - \frac{R^2}{2z^2}} + \frac{R^2/2}{z\sqrt{1 - \frac{R^2}{2z^2}}}, \tag{4}$$

and the square-shaped cloak

$$c_\diamond(z) = z\left(1 + \left(\frac{2R}{\pi z}\right)^4\right)^{3/4} \sum_{k=0}^{3} (k!)^2 \left(\frac{\sqrt{2}R}{\pi z}\right)^{4k}, \tag{5}$$

with the cut $R = i|R|$ along the imaginary axis. The choice $c(z) = u(z)$ returns the pattern to its original periodic form with a closed cloak folded to a line. The rounded square cloak is similar to eq. (5) with the sum truncated.

### Plotting the mask
The pattern is obtained by plotting the two surfaces

$$\begin{pmatrix} c(z) \\ s(z) \pm \epsilon M(u(z)) \end{pmatrix}, \tag{6}$$

into the three-dimensional space using two (black and white) colors. The real number $\epsilon > 0$ is a small number and $s(z) = (\epsilon + \frac{|z|}{|R|})$ is a Riemann sheet selection function selecting the Riemann sheet of largest value of $|z|$ to be visualized. The two surfaces are viewed from the top as a projected two-dimensional image of our pattern. Wherever $M(u(z)) > 0$, the black surface is visible. Otherwise, the white surface is visible. The parameter space $z$ is disected into $|z| > |R|$, where this map is a $1:1$ map from the region outside the circle $|z| > |R|$ to the region outside the cloak. The parameter region inside the circle $|z| < |R|$ does create several overlapping Riemann sheets and the one with largest selection function $s(z)$ is the one to be viewed. The rounded square pattern requires a more complicated selection function. There, we do not plot $s(z) \pm \epsilon M(u(z))$, if $|z| < |R|/2^{3/8}$, and if at the same time the argument $\arg(z/R) \notin [\frac{\pi}{8}, \frac{7\pi}{8}]$. The plotted pattern is sent to a company to create a mask later used for the He$^+$-ion bombardment.

### The magnetic pattern
Our thin magnetic pattern is created from a magnetic Co/Au multilayer. The multilayer has been patterned by keV He$^+$-ion bombardment through a lithographical mask[42,43] in a home-built bombardment stage[44].

### Effect of imperfections
As explained in refs. [28–30], in an ideal square pattern there are four fence points in control space. Real-world imperfections in magnetic patterns or external perturbations (external noise or imperfections in the magnetic film/disturbances in the magnetic pattern) can broaden those fence points into fence areas in control space $\mathcal{C}$. The fence area is usually a few degrees square in size and modulation loops must avoid this area to properly work as in the ideal case. Now, lets increase the size of the boat-shaped cloak like in supplementary video 2. There we have a family of fence areas instead of points, spreading at the equator and corresponding to differently rotated unit cells. In the ideal case, our control loop $\mathcal{L}_\mathcal{C}$ must pass the equator through the small gap left between the different fence families to work for all unit cells. The broadening of a fence point to an area will make this gap thinner and thus complicate the satisfaction of having the same winding number of the loop for all members of the fence family. Indeed, the larger the boat-shaped cloak, the more accurate we have to orient the control loop $\mathcal{L}_\mathcal{C}$ with respect to the pattern. Slight misalignment will destroy the proper

course of a paramagnetic particle close to the left or right boundary of the cloak. Real-world imperfections are totally unimportant for small cloaks but become a challenge for larger cloaks.

## Size limitations

Apart from rotating individual domains, the conformal map also changes the size of the domains. Let us pick the rounded square-shaped cloak to illustrate some practical challenges. Domains right at the two branch points are larger than the undeformed domains, while domains at the waist of the cloak are smaller. Larger domains can grow so large such that more than only the leading Fourier mode contributes to the colloidal potential[30]. This undermines the scalability of the problem. Smaller domains promote the formation of bipeds since individual colloidal particles are brought close to each other. In our experiments with square cloaks, we had to switch from $d = 2.8\,\mu m$ dynabeads to $d = 1\,\mu m$ myone Dynabeads to avoid the formation of bipeds at the waist.

## Energy efficiency

The robust topological control, of course, must have a prize. We have to fill a volume of the order $(cm)^3$ with our external homogeneous magnetic field in order to move a few colloidal particles of volume $(\mu m)^3$. Therefore topological control is energy inefficient as compared to other methods.

## Effect of multiparticle interactions

We eliminate the relevance of hydrodynamic interactions by the adiabatic driving of the control loop and adiabatic loop frequencies are of the order 0.1 Hz. In contrast to hydrodynamic interactions, dipolar interactions play a role in our experiments, since they assemble individual paramagnetic colloidal particles into bipeds that are rods of several paramagnetic colloids aligned along the external field direction. Some bipeds can be seen in the supplementary video 6. They fall into a different topological class than the individual colloids, and we have exploited this in various previous works[33,35,45]. They do not avoid the cloak because of this, as can be checked by watching supplementary video 6.

## Data availability

All the data supporting the findings are available from the corresponding author. All trajectories of particles are extracted from Supplementary Videos 3–5. We append the tracked particle files and a python code in a supplementary data set called Supplementary Data 1 that converts this data into Figures 3 and 4.

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

## Acknowledgements

T.M.F. and D.d.l.H. acknowledge funding by the Deutsche Forschungsgemeinschaft (DFG, German Research Foundation) under project number 531559581. D.D.I.H. acknowledges support via the Heisenberg program of the DFG via project number HE 7360/7-1. P.K., F.S., and M.U. acknowledge financial support from the National Science Center Poland through OPUS funding (Grant No. 2019/33/B/ST5/02013). SA acknowledges funding by a PhD scholarship from Kassel university. A.E., A.V., S.A. acknowledges support by the Deutsche Forschungsgemeinschaft (DFG, German Research Foundation) under project numbers INST 159/139-1 FUGG, INST 159/94-1 FUGG, and INST 159/95-1 FUGG. A.V. acknowledges support by DFG under project no. 499361641. Open Access funding enabled and organized by Projekt DEAL.

## Author contributions

A.R. performed the experiments. The Bayreuth team A.R., T.M., N.S., D.d.l.H., and T.M.F. designed the experiment. A.R. and T.M.F. wrote the manuscript with input from all the other authors. N.S. computed the cloak patterns. T.M. prepared the samples. The Poznań team P.K., F.S., and M.U., produced the magnetic film. The Kassel team S.A., A.J.V., and A.E. performed the fabrication of the micromagnetic domain patterns within the magnetic thin film.

## Funding

## Competing interests

The authors declare no competing interests.
