## [Transparent Peer Review file · Nature Communications]

Topologically cloaked magnetic colloidal transport

Corresponding Author: Professor Thomas Fischer

Version 0:

Reviewer comments:

Reviewer #1

(Remarks to the Author)

In this article, the authors extend their previous work on topologically protected transport to include impermeable regions that nevertheless maintain the same transport properties after particles have moved past. These regions are called cloaked, similar to what has previously been observed in optical and acoustic metamaterials.

The topological transport occurs in antiferromagnetic domain lattices, and the cloaked regions are deformations of the original lattice. Previously, the authors had studied the requirements in control space to produce transport, and in this article, they use these ideas to ensure there is no transport possible along the cloaked regions. When the cloaked region is introduced, it must deform the lattice around it. As long as the fence points of these deformed lattices fit inside the control space loop the lattice will keep sustaining transport. Using these results, the authors study several different shapes of the cloak to find which conditions work reliably as the cloak size is increased. They find that a good scale-free cloak is the boat shape.

Overall, the results are original and could probably be included in the current toolbox of lab on-a-chip technologies. However, the most interesting aspect is the mathematical part, and it would be interesting to think whether these results can be extended to other transport mechanisms. The conclusions are well supported by the results, and I can therefore recommend the publication in Nat. Comms.

I have a few suggestions that could help the overall clarity.

* Fence points are discussed without introduction. In fact, the whole relationship between control space and action space is not explained, but it is expected that the reader knows the previous work. I think this should be explained to make the work self-contained.

* The paragraph starting with "In Fig. 1 we have used global rotations of the pattern..." is quite impenetrable. Is there a way in which this could be presented more clearly to an interdisciplinary audience?

* Figure 4 shows the motion of all particles going through a Wigner Seitz cell, and distinguishes between trajectories that respect the topological protection, and those that don't. However, this is only approximately shown by the presence of some extra points outside of the domain walls. Wouldn't it be better to connect the dots of the trajectories to see how they cross the blue lines?

* I think the appendix, which is where the methods are described, needs to be expanded. There are many details missing, and I don't think there are enough details for the work to be reproduced.

Reviewer #2

(Remarks to the Author)

Report on the manuscript "Topologically cloaked magnetic colloidal transport" by Anna M. E. B. Rossi, Thomas M^rarker, Nico C. X. Stuhlmüller, Piotr Kuswik, Feliks Stobiecki, Maciej Urbaniak, Sapida Akhundzada, Arne J. Vereijken, Arno Ehresmann, Daniel de las Heras, and Thomas M. Fischer, submitted to Nature Communications.

Overall Summary:

The paper demonstrates topological cloaking of particles by techniques akin to coordinates transformation physics, backed by a rigorous theoretical work and a successful experimental validation. Such a novel method for cloaking unit cells in a magnetic pattern, preventing colloidal paramagnetic particles from interacting with these cloaked regions during transport, is a significant advancement in the field of colloidal transport. A demonstration is supplied, of how particles travel around cloaked regions by topologically robust paths, behaving as though the cloaked region was not there. Such a mechanism leans on the deformation of periodic magnetic patterns as well as using an external magnetic field loop to manipulate the particle transport.

The paper is a breakthrough in its field.

Overall, I recommend the publication in Nature Communications of this pioneering paper, after some revisions suggested below.

In more details:

The application of cloaking techniques to particle transport, is a novelty in the field. This is achieved through the coordinate transformation technique, via a biholomorphic mapping of a deformed magnetic pattern, allowing particles to bypass the cloaked regions without a significant alteration to their trajectory upon exiting the cloaked area. The work unveils more insights into the scalability of cloaks and some limits on the cloaking. Namely, it reveals that specific cloaking shapes like boat-shaped designs can be scaled to arbitrary sizes, while circular cloaks are prone to decloaking at some critical curvatures. For topological robustness, the maximum permissible local rotation of the pattern must then remain under 45 degrees.

The theoretical predictions are successfully validated by an experimental verification, using colloidal particles above magnetic lattices, with different shapes and sizes, such as circular, boat, and square-shaped cloaks. The work also further comes with some videos and microscopy images that illustrate the trajectories of colloidal particles traveling around the cloaked regions.

The authors discuss the practical potential applications of their work in lab-on-chip devices, which offer a way to secure chemical transport without interference by creating inaccessible areas that other transported chemicals cannot reach.

Extending cloaking from wave-based systems to dissipative particle systems is a significant innovative approach, particularly in practical contexts such as microfluidics. The authors rigorously use mathematically robust tools such as complex conformal mappings to design their cloaks. The combination of theory with experimental data, microscopy images and video documentation, together with the hint at applications in secure chemical transport and other lab-on-chip technologies, strengthen the paper's claims.

A few remarks to improve the paper:

The paper could benefit from a deeper investigation into why the decloaking mechanism under certain conditions (e.g., critical curvature for circular cloaks) occur and potential remedies. More data or comments on the transition from cloaked to decloaked states, as well as an investigation into the dynamics at the cloak's boundaries, would supply clearer insights on the boundaries of the method's applicability and practical limitations.

Besides the topological protection upon which the cloaking relies, a further analysis of the real-world imperfections in magnetic patterns or external perturbations (external noise or imperfections in the magnetic film/disturbances in the magnetic pattern) which might affect the robustness of the cloaking, would help to improve the practical insights offered by the paper and understand the level of tolerance of the cloak.

Given the novelty of cloaking particles instead of waves, the paper may also benefit from exploring possible potential impactful applications in other fields beyond microfluidics, such as biological systems, biomedicine, targeted drug delivery, material science, soft robotics, water treatment, environmental remediation, or other types of colloidal suspensions. This may enhance the relevance and appeal of the research.

The authors mathematically justify the fact that certain shapes, such as the boat-shaped cloak, can be scaled to arbitrary sizes, unlike others such as the circular ones. A word on the practical challenges related to the scaling (e.g. limitations, pattern deformation constraints) would be useful to the paper.

Another analysis that could be useful, would be that of the energy efficiency of this process, the energy cost of maintaining the external magnetic field compared to other particle transport methods.

A discussion on how significant multi-particle interactions (under, e.g. hydrodynamic forces, magnetic dipoles, aggregation effects, ...) might influence the cloaking efficiency, especially in denser particle environments (as opposed to individual or small ensembles of colloidal particles), would add depth to the study.

Conclusion

The paper addresses the novel challenge of cloaking particles and is a breakthrough in its area. The subject and the methodology are sound and the paper well written. It is a significant contribution in the field of colloidal transport, which successfully demonstrates the topological cloaking of particles, While some question in the possible practical limitations need addressing, the importance of the potential applications for microfluidics and lab-on-chip technologies are very promising. Overall, I recommend its publication in Nature Communications, after the above revisions.

Version 1:

Reviewer comments:

Reviewer #1

(Remarks to the Author)

The authors have address my concerns. The new version better highlights their important results. I can now recommend publication in Nat. Comms.

Reviewer #2

(Remarks to the Author)

No further comments.

Reviewer 1 (Remarks to the Author):

In this article, the authors extend their previous work on topologically protected transport to include impermeable regions that nevertheless maintain the same transport properties after particles have moved past. These regions are called cloaked, similar to what has previously been observed in optical and acoustic metamaterials.

The topological transport occurs in antiferromagnetic domain lattices, and the cloaked regions are deformations of the original lattice. Previously, the authors had studied the requirements in control space to produce transport, and in this article, they use these ideas to ensure there is no transport possible along the cloaked regions. When the cloaked region is introduced, it must deform the lattice around it. As long as the fence points of these deformed lattices fit inside the control space loop the lattice will keep sustaining transport. Using these results, the authors study several different shapes of the cloak to find which conditions work reliably as the cloak size is increased. They find that a good scale-free cloak is the boat shape.

Overall, the results are original and could probably be included in the current toolbox of lab on-a-chip technologies. However, the most interesting aspect is the mathematical part, and it would be interesting to think whether these results can be extended to other transport mechanisms. The conclusions are well supported by the results, and I can therefore recommend the publication in Nat. Comms.

We thank the Referee for the positive judgment, the constructive comments and the recommendation for publication. We respond to all points raised by the Referee in the following.

I have a few suggestions that could help the overall clarity.

Fence points are discussed without introduction. In fact, the whole relationship between control space and action space is not explained, but it is expected that the reader knows the previous work. I think this should be explained to make the work self-contained.

Although we do not want to repeat the entire contents of our previous papers, we agree with the referee that adding more information about essential findings is a good idea. Therefore we tried to introduce the fence points and explain the relationship between control space and action space in our revised manuscript:

Changes: Our new text marked as C2 in the highlighted version of the manuscript now reads:

There are special orientations in control space, called fence points. These orientations are the only orientations for which the position of a paramagnetic particle inside one unit cell is marginally stable. We have marked the fence points $\mathcal{F}_{\pm x, \pm y}^C$ by a superposition of colored diamonds and colored spheres. The color of the diamonds codes the two rotated

and one unrotated orientations of the pattern. For one orientation of the pattern, the color of the sphere codes the type $\pm x, \pm y$ of four different fence points. In action space (Fig. 1c)) these fence points generate fence lines $\mathcal{F}_{\pm x, \pm y}^A$ of these same four colors. In references [28-30] we have shown that per winding of the control loop \mathcal{L}_C around a fence point $\mathcal{F}_{\pm x, \pm y}^C$ in control space \mathcal{C} the colloids are transported by one unit cell on a slalom-like path along the corresponding $\mathcal{F}_{\pm x, \pm y}^A$ -lines in action space \mathcal{A} .

The paragraph starting with "In Fig. 1 we have used global rotations of the pattern..." is quite impenetrable. Is there a way in which this could be presented more clearly to an interdisciplinary audience?

We thank the referee for letting us know that this part of the manuscript is rather difficult. We distinguish global rotations, $e^{i\phi}$, with $d\phi/dz = 0$ that rotate the entire pattern by one angle ϕ from local rotations, $e^{i\phi(z)} = e^{i\arg(du/dz)}$, with $d\phi/dz \neq 0$ that rotate individual patches of the pattern by different angles. Global rotations are used to create the pattern of Fig.1, local rotations are used via the conformal map to create the other patterns.

Changes: We have corrected the paragraph toward:

In Fig. 1 we have used two global rotations that rotate the entire pattern of Fig. 1c either toward the green or the olive pattern of Fig. 1b. The two global rotations cause the formation of sharp twin boundaries (between the green and the olive patterns) that must be passed by the particles and that are the source of more complicated transport. We can avoid such domain walls using local instead of global rotations. A local rotation field preserves the shape of the pattern locally but smoothly rotates neighboring regions relative to each other.

see the paragraph marked C3

Figure 4 shows the motion of all particles going through a Wigner Seitz cell, and distinguishes between trajectories that respect the topological protection, and those that don't. However, this is only approximately shown by the presence of some extra points outside of the domain walls. Wouldn't it be better to connect the dots of the trajectories to see how they cross the blue lines?

The density of trajectory points is high in the region of the master trajectory, but low in the region that would be void if everything was perfectly cloaked. In our new Fig. 4c) we therefore plot trajectories as trajectory points in the high density region and as trajectory lines in the low density region. Points in the low density region not connected to lines correspond to trajectories leading toward the high density region via the Wigner Seitz cell boundaries and not via the blue domain walls. Note that most of the trajectory lines in Fig. 4c) in the region close to the blue domain walls arise from the square cloak, specifically from the small unit cells near the waist of the square. No matter which shape, there are no transfers of trajectories across the marked domain walls, as can be inferred for the boat shape also from Fig. 4b).

Like in our new Fig. 4c), in our new Fig. 4f) we plot the trajectories near the master curve as points and we connect the trajectory points near the blue domain walls with black trajectory lines. Note that those lines frequently cross the blue domain walls. Such crossings are absent in Fig. 4c) emphasizing the difference of cloaked and decloaked trajectories. These crossings all arise due to the transport behavior of the red trajectories near the branch points (Fig. 3) and it shows that there is only a small region where the decloaking changes the transport behavior.

We have amended the text of the manuscript explaining this in the paragraph marked C5 and C6 and we have amended the caption of Fig. 4 accordingly

I think the appendix, which is where the methods are described, needs to be expanded. There are many details missing, and I don't think there are enough details for the work to be reproduced.

We agree with the referee and have added more details to our methods section.

Changes: We have renamed the appendix into methods see C7 and we have inserted several subsections C8-C12

C8=**Plotting the mask:**

C9=**Effect of imperfections:**

C10=**Size limitations:**

C11=**Energy efficiency:**

C12=**Effect of multiparticle interactions:**

for a more in depth description of our methods used.

We thank the Referee for the constructive criticism and for the for the positive judgement.

Reviewer 2 (Remarks to the Author):

Overall Summary:

The paper demonstrates topological cloaking of particles by techniques akin to coordinates transformation physics, backed by a rigorous theoretical work and a successful experimental validation. Such a novel method for cloaking unit cells in a magnetic pattern, preventing colloidal paramagnetic particles from interacting with these cloaked regions during transport, is a significant advancement in the field of colloidal transport. A demonstration is supplied, of how particles travel around cloaked regions by topologically robust paths, behaving as though the cloaked region was not there. Such a mechanism leans on the deformation of periodic magnetic patterns as well as using an external magnetic field loop to manipulate the particle transport.

The paper is a breakthrough in its field.

Overall, I recommend the publication in *Nature Communications* of this pioneering paper, after some revisions suggested below.

In more details:

The application of cloaking techniques to particle transport, is a novelty in the field. This is achieved through the coordinate transformation technique, via a biholomorphic mapping of a deformed magnetic pattern, allowing particles to bypass the cloaked regions without a significant alteration to their trajectory upon exiting the cloaked area. The work unveils more insights into the scalability of cloaks and some limits on the cloaking. Namely, it reveals that specific cloaking shapes like boat-shaped designs can be scaled to arbitrary sizes, while circular cloaks are prone to decloaking at some critical curvatures. For topological robustness, the maximum permissible local rotation of the pattern must then remain under 45 degrees.

The theoretical predictions are successfully validated by an experimental verification, using colloidal particles above magnetic lattices, with different shapes and sizes, such as circular, boat, and square-shaped cloaks. The work also further comes with some videos and microscopy images that illustrate the trajectories of colloidal particles traveling around the cloaked regions. The authors discuss the practical potential applications of their work in lab-on-chip devices, which offer a way to secure chemical transport without interference by creating inaccessible areas that other transported chemicals cannot reach.

Extending cloaking from wave-based systems to dissipative particle systems is a significant innovative approach, particularly in practical contexts such as microfluidics. The authors rigorously use mathematically robust tools such as complex conformal mappings to design their cloaks. The combination of theory with experimental data, microscopy images and video documentation, together with the hint at applications in secure chemical transport and other lab-on-chip technologies, strengthen the paper's claims.

We are flattened by the reviewers praise of our work and we are thankful for thoroughly reviewing the manuscript, for providing constructive feedback and for the positive recommendation.

A few remarks to improve the paper:

The paper could benefit from a deeper investigation into why the decloaking mechanism under certain conditions (e.g., critical curvature for circular cloaks) occur and potential remedies. More data or comments on the transition from cloaked to decloaked states, as well as an investigation into the dynamics at the cloak's boundaries, would supply clearer insights on the boundaries of the method's applicability and practical limitations.

We agree with the referee and have amended the paper accordingly. A fence line in the undeformed u -plane passing through a branch point $du/dz = 0$ is mapped onto

two fence lines (one outside and one inside the cloak) approaching the branch point in the z plane, both making a turn by $\pm\pi/2$ and exiting the branch point at right angles compared to the incoming fence lines and along the cloak boundary. Hence the vicinity of the cloak boundary violates the requirement that the map should turn by an angle less than $|\pi/4|$. For a flat cloak boundary without curvature, the region violating the $|\arg(du/dz)| < |\pi/4|$ requirement emanates along the entire cloak boundary. The remedy to limit the region of violation is to bend the cloak boundary in the z -plane until it encloses an angle less than $|\pi/4|$ with the fence line in the u plane. The fence line through the branch point corresponds to a fence point in control space that is not wound around. We wind around the opposite fence point. Fence lines of the used fence point are at a distance of half a unit cell from the branching and unused fence line. If the curvature of the circular cloak is large enough then the region of violation of the $|\arg(du/dz)| < |\pi/4|$ requirement does not reach the used fence line and cloaking works. There is no remedy for a circular cloak of insufficient curvature. The ultimate remedy of avoiding large angles is to use a second map $c(z)$ with $du/dz = dc/dz = 0$ both vanishing simultaneously at the branch point and $|\arg(du(z)/dc(z))| < \pi/4$. That is exactly what we have done using the boat shaped cloak, and approximately what we have done with the rounded square shaped cloak. When we sharpen the edges of the rounded square shaped cloak we can recloak a decloaked large rounded square shaped cloak.

Changes: We clarify in the text marked C4 how to remedy the problems at the branch point. The text reads:

There is no cusp of the cloak shape at the branch point of the circular cloak, i.e. the unit cell at the branch point locally rotates by $\pi/2$. We achieve a cusp ($|\arg[dc(z)/du(z)]|_{z=\pm R} < \pi/4$) at the branch point with the second map $c(z)$. Like the differential of first map, the differential of the second map must vanish at the branch points ($dc/dz|_{z=\pm R} = du/dz|_{z=\pm R} = 0$). This is the case for the boat shaped and the square shaped cloak.

Besides the topological protection upon which the cloaking relies, a further analysis of the real-world imperfections in magnetic patterns or external perturbations (external noise or imperfections in the magnetic film/disturbances in the magnetic pattern) which might affect the robustness of the cloaking, would help to improve the practical insights offered by the paper and understand the level of tolerance of the cloak.

Let us answer this question for the periodic pattern first. As explained in detail in refs [28-30], in an ideal square pattern there are four fence points in control space. Real-world imperfections in magnetic patterns or external perturbations (external noise or imperfections in the magnetic film/disturbances in the magnetic pattern) broaden those fence points into fence areas in control space. The area in control space is usually a few degrees square in size and loops must avoid this area to properly work as in the ideal case. Now lets increase the size of the boat shaped cloak like in supplementary video 2. There we have a family of areas instead of points spreading at the equator and corresponding to differently rotated unit cells. In the ideal case our control loop \mathcal{L}_C must pass the equator through the small gap left between the different families to work for all

unit cells. The broadening of the fence point to an area will make the gap thinner and thus complicate the satisfaction of having the same winding number of the loop for all members of the family. Indeed the larger the boat shaped cloak the more accurate do we have to orient the control loop \mathcal{L}_c with respect to the pattern. Slight misalignment will destroy the proper course of a paramagnetic particle close to the left or right boundary of the cloak. Real-world imperfections are unimportant for small cloaks but become a challenge for larger cloaks

Changes: We have added a paragraph **Effect of imperfections:** marked C9 to the methods section.

Given the novelty of cloaking particles instead of waves, the paper may also benefit from exploring possible potential impactful applications in other fields beyond microfluidics, such as biological systems, biomedicine, targeted drug delivery, material science, soft robotics, water treatment, environmental remediation, or other types of colloidal suspensions. This may enhance the relevance and appeal of the research.

Although we agree with the referee that our work might have applications in other particle related systems, we are not aware of a concrete example. We have commented about this point at the end of the introduction.

We have added the sentence marked C1:

This is just one example of the use of cloaking for the transport of colloids. There are abundant applications of cloaking in wave like systems [1-26]. Since we show here that particle like systems can also exhibit cloaking, this opens possibilities for new applications in particle related fields.

The authors mathematically justify the fact that certain shapes, such as the boat-shaped cloak, can be scaled to arbitrary sizes, unlike others such as the circular ones. A word on the practical challenges related to the scaling (e.g. limitations, pattern deformation constraints) would be useful to the paper.

Apart from rotating individual domains, the conformal map also changes the size of the domains. Let us pick the rounded square shaped cloak to illustrate some practical challenges. Domains right at the two branch points are larger than the undeformed domains, while domains at the waist of the cloak are smaller. Larger domains can grow so large such that more than only the leading Fourier mode contribute to the colloidal potential. This undermines the scalability of the problem. Smaller domains promote the formation of bipeaks since individual colloidal particles are brought close to each other. In our experiments with square cloaks we had to switch from $d = 2.8\mu\text{m}$ dynabeads to $d = 1\mu\text{m}$ myone dynabeads to avoid the formation of bipeaks at the waist.

Changes: We have added a paragraph **Size limitations** marked C10 to the methods

section.

Another analysis that could be useful, would be that of the energy efficiency of this process, the energy cost of maintaining the external magnetic field compared to other particle transport methods.

The robust topological control of course must have a prize. We have to fill a volume of the order cm^3 with our external homogeneous magnetic field in order to move a few colloidal particles of volume $(\mu\text{m})^3$. Therefore topological control is energy inefficient as compared to other methods.

Changes: We have added a paragraph **Energy efficiency** marked C11 to the methods section.

A discussion on how significant multi-particle interactions (under, e.g. hydrodynamic forces, magnetic dipoles, aggregation effects, ...) might influence the cloaking efficiency, especially in denser particle environments (as opposed to individual or small ensembles of colloidal particles), would add depth to the study.

We eliminate the relevance of hydrodynamic interactions by the adiabatic driving of the control loop. Loop frequencies are of the order 0.1 Hz. Dipolar interactions play a role in our experiments. Dipolar interactions assemble individual paramagnetic colloidal particles into bipeds that are rods of n paramagnetic colloids aligned along the external field direction. Some bipeds can be seen in the supplementary video 6. They fall into a different topological class than the individual colloids and we have exploited this in various previous works. They do not avoid the cloak because of this as can be checked by watching supplementary video 6.

Changes: We have added a paragraph **Effect of multiparticle interactions:** marked C12 to the methods section.

Conclusion

The paper addresses the novel challenge of cloaking particles and is a breakthrough in its area. The subject and the methodology are sound and the paper well written. It is a significant contribution in the field of colloidal transport, which successfully demonstrates the topological cloaking of particles, While some question in the possible practical limitations need addressing, the importance of the potential applications for microfluidics and lab-on-chip technologies are very promising. Overall, I recommend its publication in Nature Communications, after the above revisions.

We thank all two referees for their detailed and helpful reviews of our work

We thank all two referees for the time invested in our manuscript, and for their detailed and helpful reviews of our work